# ARQ: A Mixed-Precision Quantization Framework for Accurate and Certifiably Robust DNNs

## Abstract

Mixed precision quantization has become an important technique for enabling the execution of deep neural networks (DNNs) on limited resource computing platforms. Traditional quantization methods have primarily concentrated on maintaining neural network accuracy, either ignoring the impact of quantization on the robustness of the network, or using only empirical techniques for improving robustness. In contrast, techniques for robustness certification, which can provide strong guarantees about the robustness of DNNs have not been used during quantization due to their high computation cost.

This paper introduces ARQ, an innovative mixed-precision quantization method that not only preserves the clean accuracy of the smoothed classifiers but also maintains their certified robustness. ARQ uses reinforcement learning to find accurate and robust DNN quantization, while efficiently leveraging randomized smoothing, a popular class of statistical DNN verification algorithms, to guide the search process. We compare ARQ with multiple state-of-the-art quantization techniques on several DNN architectures commonly used in quantization studies: ResNet-20 on CIFAR-10, ResNet-50 on ImageNet, and MobileNetV2 on ImageNet. We demonstrate that ARQ consistently performs better than these baselines across all the benchmarks and the input perturbation levels. In many cases, the performance of ARQ quantized networks can reach that of the original DNN with floating-point weights, but with only $1.5\%$ instructions.

## 1 Introduction

Mixed precision quantization has become an important technique for enabling the execution of deep neural networks (DNNs) on limited resource computing platforms. Quantization of an original DNN model with floating-point weights/activations significantly reduces the model size and the complexity of operations, while retaining the model's accuracy (Wang et al., 2019). However, quantizing the model also reduces its *robustness*, i.e., the ability of the network to produce a correct classification in the presence of (even small) adversarial input perturbations.

To alleviate this problem, *certified DNN robustness* techniques provide formal guarantees that the DNN will classify all small perturbations with the same label as the original input. These techniques are designed to protect against a broad scope of possible adversarial inputs, unlike commonly used *empirical robustness* techniques, which defend against only a specific kind of adversarial inputs, typically in a best-effort fashion (Raghunathan et al., 2018; Li et al., 2023).

Despite their desirability, certified robustness techniques, have a high computational cost. Many deterministic techniques that have been developed have been used only on small networks and datasets (Singh et al., 2019; Lechner et al., 2022; Zhang et al., 2022). Statistical methods for robustness certification based on *randomized smoothing* (RS) (Cohen et al., 2019) currently offer the greatest scalability. Yet, due to their cost, RS and other similar techniques have not been used *during* quantization but only at the end of the optimization to characterize the level of robustness of the final quantized network. Using these techniques during model quantization is an open question.

**Our Work: ARQ.** We present ARQ, a novel robustness-aware mixed-precision quantization framework for neural networks. ARQ demonstrates for the first time that robustness certification can be

included as a part of the search for optimal quantization of an original DNN. ARQ's algorithm takes a pre-trained DNN and encodes a reinforcement learning (RL) problem that searches for *quantization policies* – the bit-widths of the weights/activations of all layers in the DNN – that (1) preserve DNN's accuracy, (2) improve its robustness, and (3) reduce the model's computation cost. This design allows ARQ to support mixed-precision quantization (MPQ), in which the weights in each layer can be quantized with different bit-widths, thus giving fine-grained control over the possible policies.

The key insight behind ARQ's approach is that the optimization for both accuracy and robustness aims to maximize the DNN's *certified radius*, which characterizes all the slightly perturbed inputs that the DNN classifies with the same label as the non-perturbed input. This optimization objective fits well within the RL framework. ARQ's algorithm also makes it possible to leverage recent approaches for incremental analysis of certified robustness to speed up the quantization policy search.

We compare ARQ with existing searching and learning-based mixed precision quantization methods that only optimize for accuracy. The baselines include HAQ (Wang et al., 2019) (which uses reinforcement learning) and LIMPQ (Tang et al., 2023) (which uses integer programming) and fixed-precision quantization method PACT (Choi et al., 2018). We considered three DNN architectures commonly used in quantization studies: ResNet-20 on CIFAR-10, ResNet-50 on ImageNet, and MobileNetV2 on ImageNet.

We demonstrate that ARQ consistently performs better than these baselines across all the benchmarks and the input noise levels. In many cases, ARQ can even reach, and sometimes even improve on the accuracy and robustness of the original floating-point network, but with only $1.5\%$ operations.

**Contributions.** The paper makes the following contributions:

- **Approach:** We present ARQ, the first approach for mixed-precision quantization that optimizes for certified robustness of DNNs. It poses an optimization problem that maximizes the certified radius for a bounded resource usage cost (e.g., compute instructions, model size).
- **Framework:** ARQ's algorithm incorporates randomized smoothing within the reinforcement learning loop, which enables it to find certifiably robust quantized networks.
- **Results:** Our experiments on three commonly used networks/datasets show that ARQ consistently performs better than the state-of-the-art quantization techniques.

## 2 BACKGROUND

### 2.1 MIXED PRECISION QUANTIZATION

Neural network quantization is a model compression technique that can reduce a network's size and compute cost. Quantization applies to float-valued weights and activations in the network and converts them to integer values of certain bit-widths. Using the same bit-width for the entire network is sub-optimal because some layers are more amenable to quantization than others.

Mixed Precision Quantization assigns different bit-widths per weight or activation in a network and searches for the best combination of bit-widths. A *quantization policy* $P$ is a sequence of bit-width assignments to each layer in the network. For a network of $L$ layers, where each layer has $N$ bit-width options $\{b_1, b_2, \ldots, b_N\}$ for both weights and activations, there are $N^{2L}$ combinations of quantization policies. We can then formulate the process of optimizing the quantization policy for a network $\mathbf{N}$ as the following mathematical optimization problem:

$$P_{optimal} = \arg \max_{P \in \mathcal{P}} \text{Acc}(f_P(x), y) \quad \text{s.t. } \text{Cost}(f_P) < C_0 \tag{1}$$

$$\text{Acc}(f(x), y) = \frac{1}{|X|} \sum_{(x,y) \in X} \mathbf{1}(f(x) = y) \tag{2}$$

Here $\mathcal{P}$ denotes the space of all quantization policies and $P_{optimal} \in \mathcal{P}$ is the optimal policy that maximizes $\text{Acc}(f_P(x))$ on dataset $X$, the accuracy of the quantized network $f_P(x)$. $\text{Cost}(f_P)$ is the resource usage of the network, such as the model size, the number of compute bit operations or energy consumption, and $C_0$ is a user-specified bound on the resource.

**Reinforcement Learning Based Quantization.** Wang et al. (2019); Lou et al. (2020) have introduced Reinforcement Learning (RL) based approaches to search for quantization policies. One

of the RL algorithms introduced is the Deep Deterministic Policy Gradient (DDPG) algorithm (Lillicrap et al., 2019) (see Appendix A.1 for details). The DDPG agent iteratively interacts with the environment (the neural network) by observing the state $S_k$ (the configuration of the $k_{th}$ layer), taking an action $a_k$ (the quantization bit-width), and receiving a $Reward$ (the resulting accuracy).

## 2.2 CERTIFIED NEURAL NETWORK ROBUSTNESS

A classifier is considered *certifiably robust* when its predictions are guaranteed to remain consistent within a neighborhood of input $x$. Consider a classification problem from $\mathbb{R}^m$ to classes $\mathcal{Y}$. Let $f : \mathbb{R}^m \to \mathcal{Y}$ be a neural network classifier. We seek a *smoothed* classifier $g : \mathbb{R}^m \to \mathcal{Y}$, whose prediction matches that from $f$ for any input $x$ and is *constant* within some neighborhood of $x$. Randomized smoothing (Cohen et al., 2019; Yang et al., 2020; Zhang et al., 2020) provides a way to construct such a smoothed classifier $g$ from the base classifier $f$. When queried at $x$, $g$ returns the class that $f$ is most likely to return when $x$ is perturbed by Gaussian noise:

$$g(x) := \arg\max_{c \in \mathcal{Y}} \mathbb{P}(f(x + \varepsilon) = c) \quad \text{where} \quad \varepsilon \sim \mathcal{N}(0, \sigma^2 I) \tag{3}$$

Cohen et al. (2019) show that $g$'s prediction is constant within an $l_2$ ball around any input $x$. The radius of that ball, $R(x)$, is known as the *certified radius*. $\varepsilon$ is the Gaussian noise added on the input, sampled from Gaussian distribution of mean 0 and variance $\sigma^2 I$ ($I$ is the identity matrix). $\sigma$ is the noise level, a hyperparameter of the smoothed classifier $g$ independent of the input $x$. The *certified accuracy* of a classifier is defined as the probability that the classifier correctly predicts the true labels of samples $x$ for which the certified radius $R(x)$ exceeds a certain threshold $r$. The *clean accuracy* is the certified accuracy when $r = 0$.

**Theorem 1 (From Cohen et al. (2019))** *Suppose $c_A \in \mathcal{Y}$, $\underline{p_A}, \overline{p_B} \in [0, 1]$. if*

$$\mathbb{P}(f(x + \epsilon) = c_A) \geq \underline{p_A} \geq \overline{p_B} \geq \max_{c \neq c_A} \mathbb{P}(f(x + \epsilon) = c), \tag{4}$$

*then $g(x + \delta) = c_A$ for all $\delta$ satisfying $\|\delta\|_2 \leq \frac{\sigma}{2}(\Phi^{-1}(\underline{p_A}) - \Phi^{-1}(\overline{p_B}))$, where $\Phi^{-1}$ denotes the inverse of the standard Gaussian CDF.*

Computing the exact probabilities $\underline{p_A}, \overline{p_B}$ from Eqn. 4 is intractable in general. For practical applications, RS certification utilizes sampling to estimate $\underline{p_A}$ and $\overline{p_B}$ using the Clopper-Pearson method (Clopper & Pearson, 1934). If using this procedure yields $\underline{p_A} > 0.5$, then RS algorithm sets $\overline{p_B} = 1 - \underline{p_A}$ and computes the certified radius as

$$R(x) = \sigma \cdot \Phi^{-1}(\underline{p_A}) \tag{5}$$

via Theorem 1, else it returns ABSTAIN, i.e., it cannot prove the certified robustness.

# 3 ARQ APPROACH

## 3.1 PROBLEM STATEMENT

ARQ provides a mixed precision quantization method that optimizes both the robustness and accuracy of the *quantized smoothed classifier* $g_P$ from a base classifier $f$.

**Quantization Challenges.** To improve the robustness of the quantized smoothed classifier $g_P$, one could naively replace the accuracy metric in the formulation of an existing MPQ method (Eqn. 1) with an accuracy metric for the base classifier $f_P$ that uses Gaussian noise-perturbed inputs. This approach does not significantly improve the robustness of the quantized smoothed classifier $g_P$, based on the following three observations:

- By taking one perturbed sample per data point, this accuracy metric is highly affected by randomness and does not capture the robustness of $g_P$ well.
- The accuracy of base classifier $f_P$ correlates directly with the average lower bound probability ($\underline{p_A}$) as stated in Eqn. 4. However, improving the quantized base classifier $f_P$'s accuracy on samples with $\underline{p_A} < 0.5$ does not improve the quantized smoothed classifier $g_P$'s accuracy. Because for samples where $\underline{p_A} < 0.5$, the certified radius is less than zero, indicating that the

quantized smoothed classifier $g_P$ cannot provide any robustness guarantee for these samples. Consequently, these samples cannot be correctly classified by the quantized smoothed classifier $g_P$, regardless of the improvements made to the base classifiers on such inputs.

- Using accuracy as the optimization goal does not accurately reflect the robustness of the neural networks. As $R = \sigma \cdot \Phi^{-1}(\underline{p_A})$, the radius $R$ of $g_P$ has a complicated relation to $\underline{p_A}$. Therefore, optimizing only for the accuracy of the quantized base classifier $f_P$ may not accurately translate to improvements in the accuracy of the quantized smoothed classifier $g_P$. This results in a disproportionate focus on samples with smaller certified radii, while neglecting those with larger ones, due to the non-linear relationship between $R$ and $\underline{p_A}$.

**ARQ Optimization Objective.** Instead, we propose using the certified radius of smoothed classifiers to directly guide the quantization method. It is more straightforward as it uses feedback directly from the smoothed classifiers instead of the base classifiers, and it can combine the goal of optimzing both the clean accuracy and the robustness of the smoothed classifiers. We define the following optimization problem (compare Eqn. 1) to find the optimal quantization policy $P_{optimal}$:

$$P_{optimal} = \arg\max_{P \in \mathcal{P}}(\text{Average Certified Radius}) \quad \text{s.t. Cost}(f_P) < C_0 \tag{6}$$

where Average Certified Radius (ACR) is estimated as:

$$\text{ACR} = \frac{\sigma}{|X|} \sum_{(x,y) \in X} \Phi^{-1}(\underline{\mathbb{P}(f(x + \varepsilon) = y)}) \quad \forall \varepsilon \sim \mathcal{N}(0, \sigma^2 I) \tag{7}$$

The $\underline{\mathbb{P}(f_P(x + \varepsilon) = y)}$ here represents the lower bound of probability that base classifier $f$ can correctly classify input $x$ under noise $\varepsilon$. This follows from the definition of the certified radius $R(x) = \sigma \cdot \Phi^{-1}(\underline{p_A})$ for a given input $x$ and $\mathbb{P}(f(x + \epsilon) = c_A) \geq \underline{p_A}$ in Section 2.2. By averaging over inputs in the dataset $X$, we obtain the ACR, which provides a measure of the overall robustness of the classifier. Since the clean accuracy of smoothed classifiers is the percentage of samples with a certified radius greater than zero. By focusing on optimizing the Acerage Certified Radius, we can improve both the accuracy and robustness of the quantized smoothed classifiers.

Therefore, our final robustness-aware quantization problem formulation is:

$$P_{optimal} = \arg\max_{P \in \mathcal{P}} \sum_{(x,y) \in X} \Phi^{-1}(\underline{\mathbb{P}(f_P(x + \varepsilon) = y)}) \quad \text{s.t. Cost}(f_P) < C_0 \tag{8}$$

However, it is challenging to use this formulation to search exhaustively across quantization policies because calculating the certified radius, specifically obtaining $\underline{\mathbb{P}(f_P(x + \varepsilon) = y)}$, is expensive – this probability is estimated using the Clopper-Pearson method (Clopper & Pearson, 1934), and the confidence level is related to the number of samples, and may require thousands of samples even for a single image (Cohen et al., 2019). Instead, we employ a reinforcement learning (RL) agent to search for the optimal quantization policy $P_{optimal}$, which we describe next.

## 3.2 ARQ SEARCH ALGORITHM

Algorithm 1 presents the pseudocode for the ARQ algorithm, which aims to determine the optimal quantization policy for a given DNN $f$.

We first fully certify the robustness of $g$, the smoothed version of $f$ using a large number of samples $n_0$ with function FullRobustCertify, and store the average certified radius of $g$ as $ACR_{orig}$ (line 2, 2). During each iteration, our RL agent observes the $k_{th}$ layer's configuration as state $S_k$ and uses the policy network $\mu(\cdot)$ learned from the previous iterations to determine an action $a_k$ (line 5). For each layer, the agent selects two actions for the weights and the activations of that layer. The transition $(S_k, a_k, Reward, S_{k+1}, d)$ is stored in the replay buffer $D$ for training the agent's policy network (line 7). Here, $Reward$ is initially left blank, $S_{k+1}$ is the configuration of the next layer, and $d$ is the done signal indicating if it is the last layer.

After the RL agent proposes the actions for all layers, we first transform the continuous actions in list $A$ into discrete bit-widths and combine them into a quantization policy list $P_t$. Then we evaluate the resource usage of the base classifier $f_P$, which is quantized through $P_t$. If the proposed quantization

---

**Algorithm 1** ARQ Search Algorithm

---

**Inputs:** $f$: original DNN, $\sigma$: standard deviation, $X$: inputs to the DNN, $n_0$: number of Gaussian samples used for original certification, $n$: number of Gaussian samples used for quantized model certification, $n_1$: number of Gaussian samples used for fine-tuning quantized model, $C_0$: constraint bound on the quantized models, $N$: the number of iterations for search, $D$: empty replay buffer, $\mu(\cdot)$: the policy network of the agent.

1: **function** QUANTIZATION POLICY SEARCH($f, \sigma, X, n_0, n, n_1, C_0, N, \theta, \phi, D$)
2:     $g \leftarrow$ SmoothedClassifier($f, n_0$); $ACR_{orig} \leftarrow$ FullRobustCertify($g, X, \sigma$)
3:     $P_{optimal} \leftarrow \varnothing$; $Reward_{best} \leftarrow 0$; $g_{P_{optimal}} \leftarrow \varnothing$
4:     **for** $t = 1$ **to** $N$ **do**
5:         Observe the $k_{th}$ layer's state $S_k$ and select action $a_k = \text{clip}\,(\mu(S_k) + \epsilon,\, a_{\min},\, a_{\max})$, where $\epsilon \sim \mathcal{N}_t$
6:         Observe next layer's state $S_{k+1}$, and done signal $d$ to indicate whether $S_{k+1}$ is the final layer state
7:         Store transition ($S_k$, $a_k$, $Reward$, $S_{k+1}$, $d$) in replay buffer $D$ and $a_k$ to list $A$
8:         **if** $d$ is true **then**
9:             $P_t \leftarrow$ CombineActionsToPolicy($A, C_0$)
10:            $f_P \leftarrow$ Quantize($f, P_t$); $f_P \leftarrow$ FineTune($f_P, X, \sigma, n_1$)
11:            $g_P \leftarrow$ SmoothedClassifier($f_P, n$); $ACR_P \leftarrow$ IncrementalRobustCertify($g_P, X, \sigma$)
12:            $Reward_t \leftarrow ACR_P - ACR_{orig}$
13:            **if** $Reward_t > Reward_{best}$ **then**
14:                $Reward_{best} \leftarrow Reward_t$
15:                $P_{optimal} \leftarrow P_t$; $g_{P_{optimal}} \leftarrow g_P$
16:            **end if**
17:            The $Reward$ for all transitions in this iteration is set to the final $Reward_t$.
18:            Update Q-function, policy and target network. Reset the state.
19:         **end if**
20:     **end for**
21:     **return** ($P_{optimal}$, $g_{P_{optimal}}$)
22: **end function**

---

policy $P_t$ exceeds the specified resource constraint $C_0$, we will sequentially decrease the bit-width of each layer until the constraint is finally satisfied (line 9).

The function Quantize($f, P_t$) represents the quantization on $f$ to $f_P$ with quantization policy $P_t$, where the floating-point weights and activations were mapped to integers. We finetune $f_P$ for one epoch using $n_1$ inputs in dataset $X$ with Gaussian noise of size $\sigma$ to help it recover performance (line 10). Line 11 smooth $f_P$ into a quantized smoothed classifier $g_P$ with $n$ (which is $\ll n_0$) samples. Function IncrementalRobustCertify certifies the robustness of $g_P$ incrementally by reusing the information from the initial certification of $g$ with Incremental Randomized Smoothing (IRS) (Ugare et al., 2024) to obtain the ACR of $g_P$ as $ACR_P$. The RL agent's reward for all actions, $Reward_t$, is set as $ACR_P - ACR_{orig}$, using the average certified radius of $g_P$ and $g$ to guide the learning of agent (line 12, 17). After $N$ iterations, we obtain the optimal quantization policy $P_{\text{optimal}}$, with maximim average certified radius of the quantized smoothed classifier.

### 3.2.1 QUANTIZATION POLICY SEARCH

Following previous work (He et al., 2018; Wang et al., 2019), we use DDPG as our RL agent to search the bit-widths. At the $k_{\text{th}}$ layer of the base classifier $f$, the state $S_k$ of agent is:

$$S_k = (k, c_{\text{in}}, c_{\text{out}}, s_{\text{kernel}}, s_{\text{stride}}, s_{\text{feat}}, n_{\text{params}}, i_d, i_{wa}, a_{k-1}) \tag{9}$$

where $c_{\text{in}}$ and $c_{\text{out}}$ are input/output channels, $s_{\text{kernel}}$, $s_{\text{stride}}$ and $s_{\text{feat}}$ are kernel, stride and feature map sizes, $n_{\text{params}}$ is the parameter count, $i_d$ and $i_{wa}$ indicate depthwise layers and weights/activations, and $a_{k-1}$ is the previous layer's action. The first and last layers are fixed at 8-bit quantization.

We use a continuous action space with $a_{\min} = 0$ and $a_{\max} = 1$ to keep the relative order information among different actions in line 5 of Algorithm 1. Observing the state $S_k$ and using the policy network $\mu(\cdot)$, the action $a_k$ is selected for the $k_{th}$ layer, where $\epsilon \sim \mathcal{N}_t$ is a noise term added for exploration in truncated normal distribution. We then round $a_k$ into discrete bit-width $b_k$:

$$b_k = \text{round}((b_{\min} - 0.5 + a_k \times (b_{\max} - b_{\min} + 1)), \tag{10}$$

with $b_{\min}$ and $b_{\max}$ here denoting the min and max bit-width.

As described in line 7 and line 9 in Algorithm 1, the actions will first be combined into list $A$.

$$A = (a_1, a_2, \ldots, a_k, a_{k+1}, \ldots, a_d) \tag{11}$$

where $a_d$ is the final action the agent made for the last layer. When all layers has been traversed by the agent, the list $A$ is transformed into discrete bit-width form policy $P_t$ fot the $t_{th}$ iteration.

$$P_t = (b_1, b_2, \ldots, b_k, b_{k+1}, \ldots, b_d) \tag{12}$$

$P_t$ is also limited by the resource constraint $C_0$. When the given $P_t$'s resource usage exceeds $C_0$, the bit-width will be decreased sequentially from back to front.

### 3.2.2 ROBUSTNESS-AWARE POLICY SEARCH

Due to the high cost of certifying the robustness of the quantized smoothed classifier $g_P$, we perform the certification only when $d$ is true, which indicates all actions have been taken and the entire quantization policy $P_t$ come out. This avoids the frequent and expensive robust certification for each individual quantized layer. We define our reward function $Reward_t$ to be related to only the average certified radius of the smoothed classifiers (line 12):

$$Reward_t = ACR_P - ACR_{orig} \tag{13}$$

where $ACR_P$ denotes the average certified radius gained by the quantized smoothed classifier $g_P$ through the current quantization policy $P_t$, and $ACR_{orig}$ denotes the average certified radius of the original smoothed classifier $g$, which is a function of $x$ as formulated in Eqn. 7.

The experiences in the form of transitions $(S_k, a_k, Reward_t, S_{k+1}, d)$ are stored in the replay buffer $D$ to update the Q-function, policy, and target network of the DDPG agent (line 17).

We use $Reward_{best}$ to compare with $Reward_t$ and find the optimal quantization policy $P_{optimal}$, for which the corresponding $g_{P_{optimal}}$ achieves the highest ACR.

Since our optimization goal is to maximize the average certified radius of $g_P$ across the entire policy $P_t$, we set the reward for all actions across different layers in one iteration to be the same value: the final reward $Reward_t$. This ensures the reward reflects the overall effectiveness of the quantization policy rather than individual layer actions, promoting a more comprehensive optimization.

### 3.2.3 IMPLEMENTATION DETAILS

**Quantization.** We use a linear quantization method, which maps the floating-point value to discrete integer values in the range $[-c, c]$ for weights and $[0, c]$ for activations. The quantization function Quantize($\cdot$) that quantizes floating-point weight value $v$ to $b$-bit integer value $q$ can be expressed as:

$$q = \text{round}(\text{clip}(v/s, -c, c)) \times s \tag{14}$$

where $v$ is the floating-point value, and $q$ is the quantized value. $s = \frac{c}{2^{b-1}-1}$ is the scaling factor. $c$ is optimized through the KL-divergence between $q$ and $v$. In the network, each layer utilizes two distinct $c$ values for quantizing weights and activations.

**IRS for Speedup in Certification.** Due to the significant time consumption caused by both the number of iterations required for quantization policy search and the time-consuming process of certifying each iteration of the quantized neural networks, we employ Incremental Randomized Smoothing (IRS) (Ugare et al., 2024) to certify quantized smoothed classifiers more efficiently. It is known that IRS can have similar precision to re-running RS when verifying networks that have sufficient structural similarity. Our design of ARQ algorithm aims to promote this property.

## 4 EXPERIMENTAL METHODOLOGY

**Networks and Datasets.** We evaluate ARQ on CIFAR-10 (Krizhevsky et al., 2009) and ImageNet (Deng et al., 2009) datasets. We conduct all experiments on CIFAR-10 and ImageNet with 4-bit equivalent quantization, We also perform ablation studies on CIFAR-10 dataset with various quantization levels. The initial floating-point DNNs are trained with Gaussian noises of variance $\sigma^2$ on inputs. We use ResNet-20 as the base classifier for CIFAR-10, and ResNet-50 and MobileNetV2 for ImageNet. These models are chosen because they are the most commonly used classifiers in previous studies within the areas of quantization and robustness.

**Experimental Setup.** For the ResNets experiments, we use a 48-core Intel Xeon Silver 4214R CPU with two Nvidia RTX A5000 GPUs. For the MobileNetV2 experiments, we use an AMD EPYC 7763 CPU with four Nvidia A100 GPUs. ARQ is implemented in Python and uses PyTorch.

**Hyperparameters.** We use SGD with a momentum of 0.9 and a weight decay of $10^{-4}$ for model training and fine-tuning following Cohen et al. (2019). During the policy search, we fine-tune the CIFAR-10 models for one epoch with a learning rate of 0.01, and the ImageNet models on a 60,000-sample subset with a learning rate of 0.001. For fine-tuning in evaluation, for CIFAR-10, we set an initial learning rate of 0.01 and scaled it by 0.1 at epoch 5. For ImageNet, we set an initial learning rate of $10^{-3}$ and used the ReduceLROnPlateau learning rate scheduler. Fine-tuning is limited to 10 epochs, and the batch sizes are 256 for CIFAR-10 and 128 for ImageNet. A detailed description of our fine-tuning epoch choice is described in Appendix A.3. For the optimization of the DDPG agent, following Wang et al. (2019), we use ADAM (Kingma & Ba, 2017) with $\beta_1 = 0.9$ and $\beta_2 = 0.999$. The learning rate is set to be $10^{-4}$ for the actor network and $10^{-3}$ for the critic network. During exploration, truncated normal noise with an initial standard deviation of 0.5, decaying at 0.99 per episode, is applied to the actions.

**Metrics.** We use the number of bit operations (BOPs) constraint for all methods following Yao et al. (2021). BitOPs for filter k can be represented as: $\text{BOPs}(k) = b_w \cdot b_a \cdot |k| \cdot w_k \cdot h_k / s_k^2$ where $b_w$ and $b_a$ are the bitwidths for weights and activations, $|\cdot|$ denotes the number of parameters of the filter, $w_k, h_k, s_k$ are the spatial width, height, and stride of the filter.

**Robustness Certification.** For the evaluation, we use confidence parameters $\alpha = 0.001$ for the certification of the original smoothed classifier $g$. Following the setting used by RS (Cohen et al., 2019) and IRS (Ugare et al., 2024). For policy search, we use 500 validation images, $n_0 = 10000$, and $n = 500$ samples per image. For $\zeta_x$ estimation, we use $\alpha = 0.001$ and $\alpha_\zeta = 0.001$ on CIFAR-10, and $\alpha = 0.01$ and $\alpha_\zeta = 0.01$ on ImageNet.

**Evaluation.** We compare ARQ with state-of-the-art searching and learning-based mixed-precision quantization methods HAQ (Wang et al., 2019), LIMPQ (Tang et al., 2023), NIPQ (Shin et al., 2023) and HAWQ-V3 (Yao et al., 2021) and fixed-precision quantization method PACT (Choi et al., 2018). To make the baseline methods work well, we added Gaussian noise for the inputs and used the accuracy metric on the perturbed inputs as described in Section 3.1 during the policy search process for these baselines. For certifying the original smooth classifier $g$ and quantized smooth classifier $g^P$, we used RS on 500 images each with $10^6$ samples.

ARQ code is available at: https://anonymous.4open.science/r/ARQ-FE4B.

# 5 EXPERIMENTAL RESULTS

We present our main evaluation results: (1) the robustness and clean accuracy on the CIFAR-10 and ImageNet datasets; (2) the runtime of ARQ's search algorithm; and (3) selected ablation studies.

## 5.1 ROBUSTNESS AND ACCURACY EVALUATION ON CIFAR-10

On CIFAR-10, we conducted our experiments using ResNet-20 as the base classifier, with $\sigma = \{0.25, 0.5, 1.0\}$ and various BitOPs constraint settings. Figure 1 compares ARQ and baselines.

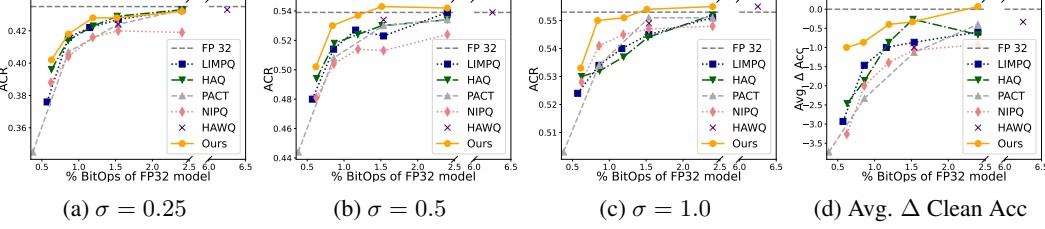

(a) $\sigma = 0.25$      (b) $\sigma = 0.5$      (c) $\sigma = 1.0$      (d) Avg. $\Delta$ Clean Acc

Figure 1: Experiments on CIFAR-10. The x-axis shows the percentage of BitOPs of $f_P$ relative to the original floating-point $f$. The y-axis shows the ACR for the first three subfigures, and the average difference in clean accuracy between the methods and the original floating-point network across different $\sigma$ settings for Figure 1d.

ARQ achieved the best ACR for all experiments except on $\sigma = 0.25$ and stricter BitOPs constraint setting, where the ACR drop compared to the best baseline is less than 0.001. Figure 1b presents

results on $\sigma = 0.50$. Notably, ARQ's 4-bit and 5-bit equivalent models outperformed the floating-point original model which was not achieved by any other methods. Figure 1c presents results on $\sigma = 1.00$. On 3-bit equivalent BitOPs constraint, ARQ had a 0.54% ACR drop while the best baseline had a 3.43% ACR drop. ARQ also outperformed the floating-point model with more than 1.5% operations.

Finally, Figure 1d shows the average clean accuracy drop achieved by the methods. ARQ's drop is smaller (hence, clean accuracy is higher) than the other baselines. Except for the 4-bit equivalent BitOPs, ARQ outperformed all baselines.

## 5.2 ROBUSTNESS AND ACCURACY EVALUATION ON IMAGENET

Table 1: Experiments on ImageNet. The ACR denotes the average certified radius, Acc denotes the clean accuracy of the smoothed classifiers, BOPs denotes the number of bit operations of the models (in G), and Size denotes the model size (in MB).

|  | Method | $\sigma = 0.25$ | | | | $\sigma = 0.50$ | | | | $\sigma = 1.00$ | | | |
|---|---|---|---|---|---|---|---|---|---|---|---|---|---|
|  |  | ACR | Acc | BOPs | Size | ACR | Acc | BOPs | Size | ACR | Acc | BOPs | Size |
| ResNet-50 | FP 32 | 0.488 | 69.4 | 4244.3 | 97.29 | 0.743 | 62.4 | 4244.3 | 97.29 | 0.914 | 45.4 | 4244.3 | 97.29 |
|  | **ARQ** | **0.472** | **70.8** | **63.32** | 13.99 | **0.724** | **61.2** | **63.50** | 13.06 | **0.916** | **46.0** | **63.50** | 12.70 |
|  | LIMPQ | 0.458 | 68.6 | 63.55 | 13.08 | 0.700 | 58.2 | 63.55 | 13.46 | 0.871 | 44.4 | 63.55 | 13.25 |
|  | HAQ | 0.460 | 69.0 | 63.56 | 13.14 | 0.715 | 60.8 | 64.15 | 13.35 | 0.880 | 44.8 | 63.55 | 11.93 |
|  | PACT | 0.460 | 69.0 | 63.56 | 13.14 | 0.715 | 61.0 | 63.56 | 13.14 | 0.884 | 45.4 | 63.56 | 13.14 |
| MobileNet-V2 | FP 32 | 0.457 | 67.0 | 308.24 | 13.24 | 0.668 | 57.0 | 308.24 | 13.24 | 0.846 | 44.4 | 308.24 | 13.24 |
|  | **ARQ** | **0.385** | **62.0** | **4.60** | 2.23 | **0.576** | **54.0** | **4.60** | 2.13 | **0.774** | **41.6** | **4.60** | 2.27 |
|  | LIMPQ | 0.347 | 58.4 | 4.62 | 2.16 | 0.540 | 50.6 | 4.62 | 2.24 | 0.703 | 40.2 | 4.62 | 2.16 |
|  | HAQ | 0.341 | 56.0 | 4.60 | 2.27 | 0.573 | 53.4 | 4.60 | 2.24 | 0.683 | 39.8 | 4.61 | 2.17 |
|  | PACT | 0.376 | 60.2 | 4.62 | 2.27 | 0.564 | 52.4 | 4.62 | 2.27 | 0.757 | 40.2 | 4.62 | 2.27 |
|  | NIPQ | 0.335 | 56.2 | 4.62 | 2.09 | 0.542 | 50.2 | 4.62 | 2.29 | 0.694 | 39.2 | 4.62 | 2.11 |

Table 1 shows the results of the experiments on ImageNet. We selected a $1.5\%$ BitOPs constraint as in CIFAR-10 experiments, it demonstrated that ARQ can achieve the accuracy and robustness of floating-point models. ARQ outperformed all other quantization methods in all settings we consider. These results show that our approach with ACR objective is able to improve *both* the clean accuracy and robustness. For ResNet-50, ACR of the ARQ-generated network is comparable to the ACR of the original floating-point model. The clean accuracy for $\sigma = 0.25$ and $\sigma = 1$ is even slightly higher than the accuracy of the original network. As a result of the limited fine-tuning, for MobileNetV2, the clean accuracy and ACR are reduced compared to the floating-point model, but both are significantly higher than the alternative quantization methods. Since low $\sigma$ certifies small radii with high accuracy but not large radii, while high $\sigma$ certifies larger radii but with lower accuracy for smaller radii, the clean accuracy drops as $\sigma$ increases. This observation is consistent with that in RS (Cohen et al., 2019).

Note, the other mixed-precision quantization methods could hardly outperform the fixed-precision quantization method PACT (Choi et al., 2018) on ImageNet, while ARQ significantly outperformed PACT on both networks and even the original floating-point model at $\sigma = 1.00$ on ResNet-50.

## 5.3 EXECUTION TIME OF ARQ SEARCH AND OTHER METHODS

Table 2 presents the time consumption for different methods. We selected $\sigma = 0.5$ as it represents the median value among the tested values in the experiments. The total time includes the policy search, fine-tuning, and evaluation. The Eval time includes the time for fine-tuning and evaluation of the quantized smooth classifiers. Although ARQ consumes more time than other methods, it is a one-time cost, and the previous section showed that it produces the most robust models.

Table 2: The total time for ARQ and other quantization search approaches (in hours).

| Benchmark | Total time | | | Policy Search | | | Eval |
|---|---|---|---|---|---|---|---|
| | ARQ | HAQ | LIMPQ | ARQ | HAQ | LIMPQ | |
| ResNet-50 | 104.75 | 75.51 | 56.60 | 71.89 | 42.65 | 23.74 | 32.86 |
| MobileNetV2 | 85.44 | 46.98 | 31.96 | 67.83 | 29.37 | 14.35 | 17.61 |
| ResNet-20 | 3.30 | 2.78 | 0.56 | 2.83 | 2.31 | 0.09 | 0.47 |

Table 3: Impact of the reward function on ResNet-20 for CIFAR-10 with $\sigma = 0.5$. The method here refers to the different approaches for the reward function.

| Method | BOPs | ACR | Radius r | | | | | | | |
|---|---|---|---|---|---|---|---|---|---|---|
| | | | 0.0 | 0.25 | 0.50 | 0.75 | 1.00 | 1.25 | 1.50 | 1.75 |
| FP 32 | 42.04 | 0.539 | 68.2 | 56.0 | 44.6 | 33.8 | 21.8 | 14.4 | 7.2 | 3.8 |
| **ARQ** | **0.354** | **0.530** | **67.2** | 54.6 | 43.2 | 32.6 | **22.2** | **14.2** | **7.4** | **4.4** |
| Val | 0.363 | 0.518 | 66.4 | 53.4 | 43.0 | **32.8** | 21.6 | 13.0 | **7.4** | 4.0 |
| Certified Acc | 0.362 | 0.525 | 65.8 | **55.6** | **43.8** | 31.8 | 21.6 | 13.0 | **7.4** | 4.0 |

Table 4: The effect of incremental RS (IRS) vs rerunning RS on CIFAR-10.

| Method | $\sigma = 0.25$ | | | $\sigma = 0.50$ | | | $\sigma = 1.00$ | | |
|---|---|---|---|---|---|---|---|---|---|
| | ACR | Acc | BOPs | ACR | Acc | BOPs | ACR | Acc | BOPs |
| FP 32 | 0.435 | 76.4 | 42.04 | 0.539 | 68.2 | 42.04 | 0.553 | 50.8 | 42.04 |
| **ARQ(-IRS)** | **0.418** | **76.0** | **0.362** | **0.530** | **67.2** | **0.354** | **0.550** | **49.6** | **0.354** |
| ARQ-RS | 0.414 | 75.8 | **0.362** | 0.526 | **67.2** | 0.359 | 0.537 | 49.2 | 0.359 |

## 5.4 ABLATION STUDIES

**Reward Function Choice.** We investigate the sensitivity of the policy search to the quality of the reward function. In ARQ, ACR is used as a reward to the RL agent. But here is an intuitive question, what if we use the certified accuracy of the quantized smooth classifier $g_P$ as the reward? As noted in Section 2.2, certified accuracy is the probability that $g_P$ correctly predicts samples $x$ with certified radius $R(x)$ exceeds the given threshold $r$. Table 3 shows the result of the ablation study conducted on 3-bit equivalent quantized ResNet-20 models. Method "Val" and "Certified Acc" refer to using the validation accuracy of $f_P$ and the certified accuracy of $g_P$ on $r = 0.5$. We observe that "Certified Acc" gains better-certified accuracy on $r = 0.25$ and $0.5$ but loses for the rest of the radii and ACR.

**The Effect of IRS.** Table 4 shows the results of using RS instead of IRS to obtain $ACR_p$ and $Reward_t$. In our experiments, IRS is 1.32x faster than RS, consistent with the results from IRS (Ugare et al., 2024). The method "RS" refers to using the same time for certification in the policy search process as the IRS. We observe that using IRS does not reduce the quality of the quantization policy (and in many cases improves it), justifying its use in ARQ's search loop as the candidate quantized networks are similar enough to benefit from incremental robustness proving.

**Other Ablation Studies.** Appendix A.2 presents quantization policies across different $\sigma$s. Appendix A.3 presents the effect of the number of epochs in fine-tuning.

## 6 RELATED WORK

**Mixed-Precision Quantization.** To optimize the balance between the accuracy and efficiency of DNNs, many mixed-precision quantization methods have been presented. Dong et al. (2019); Louizos et al. (2017); Chen et al. (2021); Tang et al. (2023) employed appropriate proxy metrics that indicate model sensitivity to quantization to generate quantization policies. Some other researchers formulated quantization policy optimization as a search problem and addressed it using a Markov Decision Process through reinforcement learning (Wang et al., 2019; Lou et al., 2020; Elthakeb

Table 5: Comparison of various robustness-aware model reduction methods. ***Model Reduction Method:*** P – Pruning; Q – Quantization; MPQ – Mixed-Precision Quantization. ***Stage:*** T – Training; PT – Post-training (tuning). ***Scale:*** Largest supported data-set.

| Approach | Properties | | | | | |
| --- | --- | --- | --- | --- | --- | --- |
| | Empirical robustness | RS | Other determ. approach | Approx. Method | Stage | Scale |
| ARQ (this work) | | ✓ | | MPQ | PT | ImageNet |
| ATMC Gui et al. (2019b) | ✓ | | | Q | T | CIFAR-100 |
| DQ Lin et al. (2019b) | ✓ | | | Q | T | CIFAR-10 |
| GRQR Alizadeh et al. (2020) | ✓ | | | Q | PT | ImageNet |
| ICR Lin et al. (2021) | | ✓ | | Q | T | Caltech-101 |
| QIBP Lechner et al. (2022) | | | ✓ | Q | T | CIFAR-10 |
| ARMC Ye et al. (2021) | ✓ | | | P | T | CIFAR-10 |
| QUANOS Panda (2020) | ✓ | | | MPQ | PT | CIFAR-100 |
| Stochastic-Shield Qendro et al. (2021) | ✓ | | | Q | PT | CIFAR-10 |
| HYDRA Sehwag et al. (2020) | ✓ | ✓* | ✓* | P | PT | ImageNet |
| TCR Sehwag et al. (2019) | ✓ | | | P | PT | CIFAR-10 |
| DNR Kundu et al. (2020) | ✓ | | | P | T | Tiny-ImageNet |
| HMBDT Giacobbe et al. (2020) | | | ✓ | Q | PT | MNIST |
| TCMR Weng et al. (2020) | | | ✓ | Q | PT | CIFAR-10 |

et al., 2020) and a differentiable search process employed Neural Architecture Search algorithms (Wu et al., 2018; Guo et al., 2020). As Table 1 shows, ARQ outperformed HAQ (Wang et al., 2019) and LIMPQ (Tang et al., 2023), which are state-of-the-art mixed-precision quantization methods.

**Robustness of Quantized Models.** As illustrated in Table 5, ARQ stands out by being the only approach that combines mixed-precision quantization with post-training optimization to achieve certified robustness on large-scale datasets like ImageNet. This distinguishes our work from other approaches that either focus only on pruning (which removes over 90% of the weights) and often target smaller datasets. Although ICR (Lin et al., 2021) has explored quantization methods with RS, ARQ focuses on post-training optimization instead of quantization-aware training and we demonstrated it can scale to ImageNet. HYDRA (Sehwag et al., 2020) analyzes pruning methods with RS but only scales to CIFAR-10 for RS and is hard to transfer to quantization methods.

Several methods were introduced to address the complementary challenge of training DNNs on security-critical and resource-limited applications. Gui et al. (2019a) unified various existing compression techniques. Lin et al. (2019a) and Alizadeh et al. (2020) presented how controlling the magnitude of adversarial gradients can be used to construct a defensive quantization method. Finally, empirical robustness approaches can improve best-effort robustness only to some kinds of adversarial inputs (Raghunathan et al., 2018; Li et al., 2023). Ugare et al. (2023; 2022) focuses on fast incremental certification with deterministic techniques but not on optimizing the DNN.

## 7 CONCLUSION AND LIMITATIONS

**Conclusion.** We introduce ARQ, the first mixed-precision quantization framework that optimizes both DNN's accuracy and certified robustness by limiting the computational resource budget. By using direct feedback from the ACR of the quantized smoothed classifier, ARQ more effectively searches for the optimal quantization policy. Our experiments demonstrate that ARQ consistently outperforms state-of-the-art quantization methods, often reaching or improving the accuracy and robustness of the original FP32 networks with down to 0.84% operations. ARQ significantly reduces the computational resource requirements of randomized smoothing, making it possible to deploy it in resource-constrained environments while providing certifiable robustness guarantees.

**Limitations.** We showed that ARQ can achieve well-quantized DNNs that match or even surpass the accuracy and robustness of the original DNNs with floating-point weights. However, these properties are still dependent on the training of the original network, which should be trained with Gaussian augmentation to ensure the quantized network performs well. Deploying DNNs with mixed-precision inference can be more challenging compared to the fixed-precision methods, however, recent works aim to address this issue (Sharma et al., 2018). The current implementation of ARQ has been evaluated solely on image classification tasks. In the future, we plan to investigate how ARQ performs on other complex tasks such as object detection and natural language processing.

REPRODUCIBILITY STATEMENT

We provide the source code and configuration details necessary to reproduce our experimental results. Detailed descriptions of the hyperparameters used in our experiments are provided in Section 4, and the pseudocode for ARQ algorithms is described comprehensively in Section 3.2. The code is available at the anonymous GitHub link provided at the end of Section 4.

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

# A APPENDIX

## A.1 THE DEEP DETERMINISTIC POLICY GRADIENT (DDPG) ALGORITHM

DDPG learns a Q-function and a policy concurrently. It uses off-policy data and the Bellman equation to learn the Q-function, and then uses the Q-function to learn the policy.

Algorithm 2 (Achiam, 2018) shows the DDPG algorithm. In some RL search-based mixed-precision quantization methods, DDPG is utilized to search for the optimal quantization policy. The environment is usually set to be the DNN itself, state $s$ is usually set to be the configuration of one layer in the DNN, the action $a$ is the continuous value that can be transformed into the bit-width for the layer, and $r$ is the reward set to be the accuracy of the DNN and computed only after all actions have been taken. The reward for all actions in one episode is set to the final accuracy gained. The agent updates only when all actions have been taken and the reward is obtained. The experiences are stored in replay buffer $D$ and randomly sampled in batch $B$ for updating the Q-function, policy network, and the target network of the agent.

---

**Algorithm 2** Deep Deterministic Policy Gradient

---

**Inputs:** Initial policy parameters $\theta$, Q-function parameters $\phi$, empty replay buffer $D$

1: Set target parameters equal to main parameters: $\theta_{\text{targ}} \leftarrow \theta$, $\phi_{\text{targ}} \leftarrow \phi$
2: **repeat**
3:      Observe state $s$ and select action $a = \text{clip}\left(\mu_\theta(s) + \epsilon, a_{\text{low}}, a_{\text{high}}\right)$, where $\epsilon \sim \mathcal{N}(0, \sigma^2)$
4:      Execute $a$ in the environment
5:      Observe next state $s'$, reward $r$, and done signal $d$ to indicate whether $s'$ is terminal
6:      Store transition $(s, a, r, s', d)$ in replay buffer $D$
7:      Set the environment state to $s'$. If $d$ is true, reset the environment state
8:      The reward for all layers is set to be the final reward
9:      **if** it's time to update **then**
10:          **for** each update step **do**
11:              Randomly sample a batch of transitions $B = \{(s_i, a_i, r_i, s'_i, d_i)\}$ from $D$
12:              Compute targets for each transition:

$$y_i = r_i + (1 - d_i)\, \gamma\, Q_{\phi_{\text{targ}}}\left(s'_i, \mu_{\theta_{\text{targ}}}(s'_i)\right)$$

13:              Update Q-function by one step of gradient descent using:

$$\phi \leftarrow \phi - \lambda_Q \nabla_\phi \left(\frac{1}{|B|} \sum_{i \in B} \left(Q_\phi(s_i, a_i) - y_i\right)^2\right)$$

14:              Update policy by one step of gradient ascent using:

$$\theta \leftarrow \theta + \lambda_\mu \frac{1}{|B|} \sum_{i \in B} \nabla_\theta \mu_\theta(s_i) \nabla_a Q_\phi(s_i, a)\big|_{a = \mu_\theta(s_i)}$$

15:              Update target networks with:

$$\theta_{\text{targ}} \leftarrow \tau\, \theta + (1 - \tau)\, \theta_{\text{targ}}$$

$$\phi_{\text{targ}} \leftarrow \tau\, \phi + (1 - \tau)\, \phi_{\text{targ}}$$

16:          **end for**
17:      **end if**
18: **until** convergence

---

## A.2 ABLATION STUDY: THE QUANTIZATION POLICIES FOR DIFFERENT LAYERS

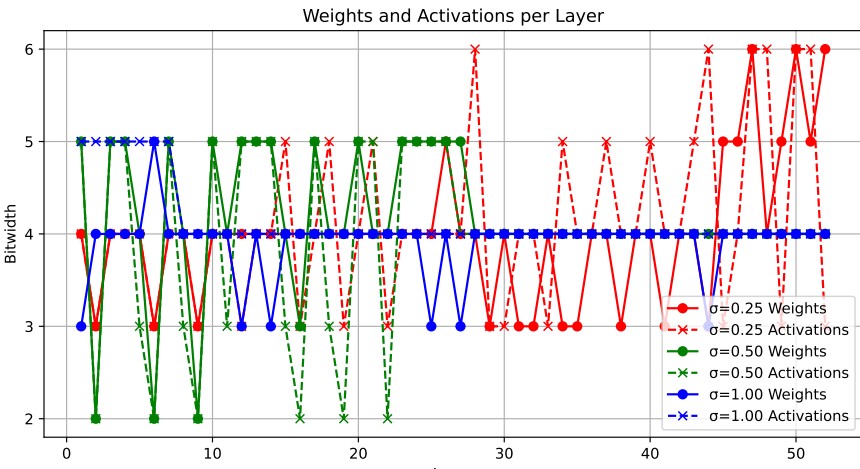

Figure 2: Quantization policy among different $\sigma$s for ResNet-50 on ImageNet. The x-axis represents the layer index, and the y-axis represents the bit-width selection in the quantization policy for each specific layer. The $\bullet$ symbol represents the bit-widths for weights, and the $\times$ symbol represents the bit-widths for activations.

Figure 2 illustrates the significance of generating different quantization policies for different values of $\sigma$. The quantization policy for $\sigma = 0.25$ is more aggressive in the first half of the layers and more conservative in the later layers. This pattern is different from that of $\sigma = 0.5$ and $\sigma = 1.0$, where the policy tends to make more adjustments in the first half and remains stable in the latter half.

## A.3 ABLATION STUDY: THE EFFECT OF THE NUMBER OF EPOCHS IN FINE-TUNING

Due to the limitation of computational resources, we only conducted fine-tuning for 10 epochs in our experiments. Table 6 shows the results of further fine-tuning for a total of 90 epochs. The trends in ACR and clean accuracy followed the same pattern as observed during the initial 10 epochs, demonstrating the effectiveness of our work.

Table 6: Experiments for ResNet-20 on CIFAR-10. The Epochs here indicates the number of epochs used for fine-tuning. The rest of Formats are similar to Table 1

| Method | GBitOPs | Epochs = 90 | | Epochs = 10 | |
|---|---|---|---|---|---|
| | | ACR | Acc | ACR | Acc |
| **ARQ** | **0.354** | **0.545** | **67.6** | **0.530** | **67.2** |
| LIMPQ | 0.361 | 0.535 | 67.0 | 0.514 | 65.0 |
| HAQ | 0.365 | 0.534 | 66.8 | 0.518 | 66.4 |
| PACT | 0.362 | 0.524 | 66.6 | 0.508 | 65.8 |

## A.4 MODEL SIZE ANALYSIS IN CIFAR-10 EXPERIMENTS

In our CIFAR-10 experiments, we used ResNet-20 as the base classifier with $\sigma = \{0.25, 0.5, 1.0\}$ and various BitOPs constraint settings. While our primary experiments focused on using BitOPs as the constraint, here we present supplementary results with ACR and average difference in clean accuracy as the y-axis and model size as the x-axis. This analysis provides additional insights

into model size measurements, even though model size was not used as a constraint in our main experiments. Figure 3 compares ARQ with baseline methods.

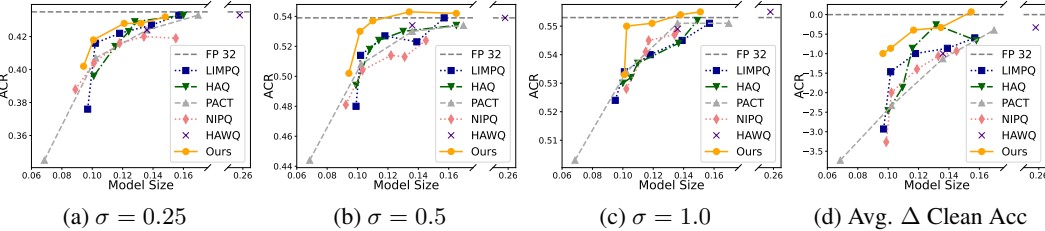

(a) $\sigma = 0.25$      (b) $\sigma = 0.5$      (c) $\sigma = 1.0$      (d) Avg. $\Delta$ Clean Acc

Figure 3: Experiments on CIFAR-10. The x-axis shows the model size of $f_P$. The y-axis shows the ACR for the first three subfigures, and the average difference in clean accuracy between the methods and the original FP32 network across different $\sigma$ settings for Figure 1d.

## A.5 ABLATION STUDY: FURTHER ANALYSIS ON THE EFFECT OF TIME CONSUMPTION IN POLICY SEARCH

Table 7: Performance of ARQ across different policy search time, showing corresponding clean accuracy, ACR, and BOPs.

| Time | Accuracy | ACR | BOPs |
|------|----------|-------|-------|
| 3.30 | 67.2 | 0.530 | 0.354 |
| 2.65 | 66.8 | 0.528 | 0.354 |
| 0.72 | 66.2 | 0.521 | 0.354 |

Table 7 presents the results of ARQ under different time constraints for policy search. This information serves as a reference for users aiming to balance time consumption in policy optimization with accuracy and robustness during inference. Notably, although performance declines with reduced search time, ARQ still outperforms the baselines.

## A.6 COMPARSION EXPERIMENTS WITH ROBUST-AWARE QUANTIZATION METHODS

Table 8: Comparision experiment with RS-based quantization method ICR on ResNet-20 for CIFAR-10 with $\sigma = 0.5$.

| Method | BOPs | Radius r | | | | | | |
|--------|-------|------|------|------|------|------|------|------|
| | | 0.0 | 0.25 | 0.50 | 0.75 | 1.00 | 1.25 | 1.50 |
| **ARQ** | **0.354** | **67.2** | **54.6** | **43.2** | **32.6** | **22.2** | 14.2 | 7.4 |
| ICR | 2.596 | 63.0 | 52.0 | 39.0 | 29.0 | 22.0 | **15.0** | **8.0** |

Table 8 showed the results comparing ARQ with ICR Lin et al. (2021). Our experiments show that ARQ 3-bit equivalent model can outperform ICR's 8-bit equivalent model in both clean accuracy and robust accuracy for certified radii below 1.0, and on radius r=1.25 and 1.50, the performance loss of ARQ is minimal considering the BOPs difference and the improvement gained on smaller radii.

Table 9 showed the results comparing ARQ with ATMC Gui et al. (2019b). We performed additional experiments on an ATMC's 8-bit equivalent model. It's shown that empirical robust quantization methods can hardly gain certified robustness.

864
865
866
867
868
869
870
871
872
873
874
875
876
877
878
879
880
881
882
883
884
885
886
887
888
889
890
891
892
893
894
895
896
897
898
899
900
901
902
903
904
905
906
907
908
909
910
911
912
913
914
915
916
917

Table 9: Comparison of certified robustness of empirical robust quantization method ATMC on ResNet-20 for CIFAR-10, showing ACR and BOPs across different $\sigma$ settings.

| Method | $\sigma = 0.25$ | | $\sigma = 0.50$ | | $\sigma = 1.00$ | |
|---|---|---|---|---|---|---|
| | ACR | BOPs | ACR | BOPs | ACR | BOPs |
| **ARQ** | **0.418** | **0.362** | **0.530** | **0.354** | **0.550** | **0.354** |
| ATMC | 0.031 | 2.596 | 0.072 | 2.596 | 0.159 | 2.596 |

