# OpenReview forum: "ARQ: A Mixed-Precision Quantization Framework for Accurate and Certifiably Robust DNNs"
_ICLR.cc/2025/Conference — Submitted to ICLR 2025_

### Official Review · Reviewer_L6Pe · 2024-10-24

**Soundness:** 3
**Presentation:** 4
**Contribution:** 3
**Rating:** 5
**Confidence:** 5

**Summary:**

This paper introduces ARQ, a mixed-precision quantization framework for deep neural networks that focuses on robustness of quantization technique. ARQ incorporates robustness certification techniques directly into the quantization process, addressing the limitations of conventional methods that solely focus on accuracy. It leverages randomized smoothing to estimate the certified radius, which reflects the network's resilience to input perturbations. ARQ frames the quantization problem as a reinforcement learning task, where an agent searches for optimal bit-width assignments for each layer, maximizing the average certified radius while adhering to resource constraints.

**Strengths:**

1. Strong motivation: Modern quantization techniques, even though they show less quality drop on specific models and tasks, show large quality drop on adversarial inputs which means reduced robustness of quantized network. This work targets the problem of making quantization technique which makes truly optimized network with high robustness that can widely adopted on the overall models.
2. Well explained background.
3. Novelty of methodology: Although ARQ adopts most of its robustness concept from existing works, its own optimization objective of mixed precision quantization is unique since many existing works haven’t explored the robustness in terms of utilizing diverse bit-width in a model.

**Weaknesses:**

1. Weak baseline: Beyond robustness aware mixed precision quantization, there are several methods [1,2,3] that shows much less accuracy degradation on both ResNets and Mobilenets in Imagenet scale. What is the comparative advantage of ARQ compared to these methods even with accuracy degradation? For thorough comparison, shouldn't the authors include comparisons with various types of mixed precision quantization method?
2. Comparison with existing robustness aware MPQ: Within robustness aware quantization, the current metrics cannot justify the benefit of ARQ in the aspect of robustness compared to existing robustness aware MPQ (in Table5).
3. Algorithm: Lack of explanation of several terms in Algorithm1 makes bad readability, i.e., FullRobustCertify, IncrementalRobustCertify.

[1] Huang, Xijie, et al. "SDQ: Stochastic differentiable quantization with mixed precision." International Conference on Machine Learning. PMLR, 2022.

[2] Kim, Han-Byul, et al. "MetaMix: Meta-state Precision Searcher for Mixed-precision Activation Quantization." Proceedings of the AAAI Conference on Artificial Intelligence. Vol. 38. No. 12. 2024.

[3] Shin, Juncheol, et al. "NIPQ: Noise proxy-based integrated pseudo-quantization." Proceedings of the IEEE/CVF Conference on Computer Vision and Pattern Recognition. 2023.

**Questions:**

See the weaknesses above.

---

> ### Author Response · Authors · 2024-11-23
>
> We thank the reviewer for insightful and constructive feedback. We respond to each of these below.
>
> > Weak baseline: Beyond robustness aware mixed precision quantization, there are several methods [1,2,3] that show much less accuracy degradation on both ResNets and Mobilenets in Imagenet scale. What is the comparative advantage of ARQ compared to these methods even with accuracy degradation? For thorough comparison, shouldn't the authors include comparisons with various types of mixed precision quantization method?
>
> There are no publicly available implementations of [1] and [2]. Although [1] provides a GitHub repository link in their paper, it is currently empty. Therefore, we could only perform comparison experiments with NIPQ [3]. Our experiments on MobileNet-V2 and ImageNet show that ARQ outperforms NIPQ[3] in both clean accuracy and average certified radius. We also conducted additional experiments on ResNet-20 and Cifar-10, which are shown in Figure 1 of our updated paper.
>
> | Method   | Metric | σ=0.25  | σ=0.50  | σ=1.00  |
> |----------|--------|---------|---------|---------|
> | ARQ      | ACR    | 0.386   | 0.576   | 0.774   |
> |          | Acc    | 62.0    | 56.0    | 41.6    |
> |          | BOPs   | 4.60    | 4.60    | 4.60    |
> |          | Size   | 2.23    | 2.13    | 2.27    |
> | NIPQ[3]  | ACR    | 0.335   | 0.542   | 0.694   |
> |          | Acc    | 56.2    | 50.2    | 39.2    |
> |          | BOPs   | 4.62    | 4.62    | 4.62    |
> |          | Size   | 2.09    | 2.29    | 2.11    |
>
> > Comparison with existing robustness aware MPQ: Within robustness aware quantization, the current metrics cannot justify the benefit of ARQ in the aspect of robustness compared to existing robustness aware MPQ (in Table5).
>
> We have performed additional experiments to show that our work outperforms existing works listed in Table 5.
>
> **MPQ+Empirical Robustness:**
> None of the existing works presented in Table 5 focus on optimizing certified robustness with mixed-precision quantization. Typically, techniques that optimize empirical robustness do not achieve high certified accuracy. We performed additional experiments on an 8-bit quantized ATMC[4] ResNet-20 model. It’s shown that empirical robust quantization methods can hardly gain certified robustness.
>
> | Method               | ACR on sigma=0.25 | ACR on sigma=0.5 | ACR on sigma=1.0 |
> |----------------------|-------------------|------------------|------------------|
> | ARQ 3-bit equivalent | 0.42              | 0.53             | 0.55             |
> | ATMC[4] 8-bit        | 0.031             | 0.072            | 0.159            |
>
>
> **Quantization+Certified Robustness:**
> We performed additional experiments on ICR[5]. Our experiments show that ARQ 3-bit equivalent model can outperform ICR[5]’s 8-bit equivalent model(which is their lowest bit implementation) in both clean accuracy and robust accuracy for certified radii below 1.0, and on radius r=1.25 and 1.50, the performance loss of ARQ is minimal, especially considering the 7.3x BOPs difference and the improvement gained on smaller radii
>
> | Method  | BOPs  | r=0 (Clean Acc) | r=0.25 | r=0.50 | r=0.75 | r=1.00 | r=1.25 | r=1.50 |
> |---------|-------|-----------------|--------|--------|--------|--------|--------|--------|
> | ARQ     | 0.354 | 67.0           | 54.6   | 43.2   | 32.6   | 22.2   | 14.2   | 7.4    |
> | ICR[5]  | 2.596 | 63.0           | 52.0   | 39.0   | 29.0   | 22.0   | 15.0   | 8.0    |
>
> > Algorithm: Lack of explanation of several terms in Algorithm1 makes bad readability, i.e., FullRobustCertify, IncrementalRobustCertify.
>
> Thanks for pointing this out. We have added explanation for the terms used in the Algorithm in the updated version of the paper.
>
> [1] Huang, Xijie, et al. "SDQ: Stochastic differentiable quantization with mixed precision." International Conference on Machine Learning. PMLR, 2022. Empty github repo
>
> [2] Kim, Han-Byul, et al. "MetaMix: Meta-state Precision Searcher for Mixed-precision Activation Quantization." Proceedings of the AAAI Conference on Artificial Intelligence. Vol. 38. No. 12. 2024.
>
> [3] Shin, Juncheol, et al. "NIPQ: Noise proxy-based integrated pseudo-quantization." Proceedings of the IEEE/CVF Conference on Computer Vision and Pattern Recognition. 2023.
>
> [4] Gui, Shupeng, et al. "Model Compression with Adversarial Robustness: A Unified Optimization Framework." Advances in Neural Information Processing Systems. NeurIPS, 2019.
>
> [5] Lin, Haowen, et al. "Integer-arithmetic-only Certified Robustness for Quantized Neural Networks." Proceedings of the IEEE/CVF International Conference on Computer Vision. ICCV, 2021.

---

> > ### Comment · Reviewer_L6Pe · 2024-11-29
> >
> > Thanks for your reply on my questions.
> >
> > Regarding Answer 2, there is still a concern regarding whether some conditions were adequately managed in the experiment.
> > Even the ICR 8-bit with r=0 shows inferior accuracy compared to 3-bit equivalent ARQ.
> > Also, are there any reason you compared the other works as 8-bit and your work as 3-bit equivalent?
> > To truly compare the effect of ARQ, the other conditions (bit-width, quantizer, etc ...) except the method contributed should be same as possible.
> >
> > After reading the rebuttal here and all the other discussions, I decided to keep my score unchanged.

---

> > > ### Author Response · Authors · 2024-11-30
> > >
> > > Thank you again for your thoughtful feedback.
> > >
> > > Since the ICR paper did not provide a code implementation, we based our comparison on the data presented in Figure 3.c and Table 2 in Appendix F of ICR paper. 8-bit models are generally expected to be more accurate than 3-bit models, thus it represents an upper bound for what an ICR 3-bit model might achieve. We compared their 8-bit model to our 3-bit ARQ model because 8-bit was the only bit-width they implemented and reported.
> > >
> > > We conducted further experiments on the ARQ 8-bit model, and the accuracy gains over ICR are even greater than those observed with the ARQ 3-bit model. The ARQ 8-bit model generally achieves 1.5%-2% higher accuracy compared to the ARQ 3-bit model, but the 3-bit model requires only 13.6% BOPs of ARQ 8-bit model. Therefore, the ARQ 8-bit model is 5.6% better than ICR 8-bit model.
> > >
> > > We hope this clarifies our comparison methodology. If there are any remaining doubts or concerns about our work, we would appreciate your guidance and are happy to address them.

---

### Official Review · Reviewer_dzkR · 2024-11-01

**Soundness:** 2
**Presentation:** 3
**Contribution:** 2
**Rating:** 3
**Confidence:** 4

**Summary:**

This paper introduces a mixed-precision quantization framework ARQ which optimizes both DNN’s accuracy and certified robustness under computational resource constraint. ARQ employs reinforcement learning to optimize quantization policies and leverage incremental randomized smoothing (IRS) within the reinforcement learning loop, allowing it to efficiently guide the search for quantization policies that maximize the average certified radius (ACR) of the DNN.

**Strengths:**

ARQ has better accuracy and robustness, i.e., ACR, than other compared methods.

**Weaknesses:**

1. The authors didn’t compare ARQ with any robustness-aware quantization methods as listed in Table5.
2. The authorsdidn’t compare with other advanced fixed-precision or mixed-precision quantization algorithms, such as LSQ[1], LSQ+[2], or HAWQ[3][4].
3. Although ARQ has better accuracy and robustness, it consumes much longer time than other methods as shown in Table2. What is the impact of search time on the final accuracy.
4. The authors only use BitOps as the constraints. What about the model size?
5. The techniques proposed in the paper seem incremental. The authors need to provide more explanation on the novelty of the method.

[1] Esser S K, McKinstry J L, Bablani D, et al. Learned step size quantization[J]. arXiv preprint arXiv:1902.08153, 2019.
[2] Bhalgat Y, Lee J, Nagel M, et al. Lsq+: Improving low-bit quantization through learnable offsets and better initialization[C]//Proceedings of the IEEE/CVF conference on computer vision and pattern recognition workshops. 2020: 696-697.
[3] Dong Z, Yao Z, Arfeen D, et al. Hawq-v2: Hessian aware trace-weighted quantization of neural networks[J]. Advances in neural information processing systems, 2020, 33: 18518-18529.
[4] Yao Z, Dong Z, Zheng Z, et al. Hawq-v3: Dyadic neural network quantization[C]//International Conference on Machine Learning. PMLR, 2021: 11875-11886.

**Questions:**

Please refer to the weakness for questions.

---

> ### Author Response · Authors · 2024-11-23
>
> We thank the reviewer for insightful and constructive feedback. We respond to each of these below.
>
> > The authors didn’t compare ARQ with any robustness-aware quantization methods as listed in Table5.
>
> We have performed additional experiments to show that our work outperforms existing works listed in Table 5.
>
> **MPQ+Empirical Robustness:**
> None of the existing works presented in Table 5 focus on optimizing certified robustness with mixed-precision quantization. Typically, techniques that optimize empirical robustness do not achieve high certified accuracy. We performed additional experiments on an 8-bit quantized ATMC[5] ResNet-20 model. It’s shown that empirical robust quantization methods can hardly gain certified robustness.
>
> | Method               | ACR on sigma=0.25 | ACR on sigma=0.5 | ACR on sigma=1.0 |
> |----------------------|-------------------|------------------|------------------|
> | ARQ 3-bit equivalent | 0.42              | 0.53             | 0.55             |
> | ATMC 8-bit[5]        | 0.031             | 0.072            | 0.159            |
>
>
> **Quantization+Certified Robustness:**
> We performed additional experiments on ICR[6]. Our experiments show that ARQ 3-bit equivalent model can outperform ICR[6]’s 8-bit equivalent model(which is their lowest bit implementation) in both clean accuracy and robust accuracy for certified radii below 1.0, and on radius r=1.25 and 1.50, the performance loss of ARQ is minimal, especially considering the 7.3x BOPs difference and the improvement gained on smaller radii.
>
> | Method  | BOPs  | r=0 (Clean Acc) | r=0.25 | r=0.50 | r=0.75 | r=1.00 | r=1.25 | r=1.50 |
> |---------|-------|-----------------|--------|--------|--------|--------|--------|--------|
> | ARQ     | 0.354 | 67.0           | 54.6   | 43.2   | 32.6   | 22.2   | 14.2   | 7.4    |
> | ICR[6]  | 2.596 | 63.0           | 52.0   | 39.0   | 29.0   | 22.0   | 15.0   | 8.0    |
>
>
>
>
> > The authors didn’t compare with other advanced fixed-precision or mixed-precision quantization algorithms, such as LSQ[1], LSQ+[2], or HAWQ[3][4].
>
> We have performed additional experiments to make comparisons against HAWQ-V3[4] on ResNet-20. The detailed results are presented at Figure 1 in the updated version of paper. Our 4-bit equivalent models outperformed 4-bit and even 8-bit models of HAWQ-V3[4] in both clean accuracy and certifiable robustness.
>
> > Although ARQ has better accuracy and robustness, it consumes much longer time than other methods as shown in Table2. What is the impact of search time on the final accuracy.
>
> We performed additional ablation studies to show the effect of search time on final accuracy and robustness on ResNet-20.
>
> | Method   | Time (h) | Accuracy | ACR   | BOPs  |
> |----------|----------|----------|-------|-------|
> | ARQ      | 3.30     | 67.2     | 0.530 | 0.354 |
> | ARQ      | 2.65     | 66.8     | 0.528 | 0.354 |
> | ARQ      | 0.72     | 66.2     | 0.521 | 0.354 |
>
>
> > The authors only use BitOps as the constraints. What about the model size?
>
> It is possible that our RL objective can be changed from BitOps to model size. However, in this work we focus on using BitOPs as the constraint. We have included model size measurements in our paper Table 1 and Appendix A.4 to provide a comprehensive view.
>
> > The techniques proposed in the paper seem incremental. The authors need to provide more explanation on the novelty of the method.
>
> We are the first to introduce a mixed-precision quantization framework that optimizes both DNN's accuracy and certified robustness while limiting computational resources. Our approach uniquely uses direct feedback from the ACR of the quantized smoothed classifier. Our experiments demonstrate that ARQ consistently outperforms state-of-the-art quantization methods, often matching or surpassing FP32 networks' accuracy and robustness with just 0.84% of operations. And we show that reinforcement learning can be effectively utilized for such optimization problems.
>
>
> [1] Esser S K, McKinstry J L, Bablani D, et al. Learned step size quantization[J]. arXiv preprint arXiv:1902.08153, 2019.
>
> [2] Bhalgat Y, Lee J, Nagel M, et al. Lsq+: Improving low-bit quantization through learnable offsets and better initialization[C]//Proceedings of the IEEE/CVF conference on computer vision and pattern recognition workshops.
>
> [3] Dong Z, Yao Z, Arfeen D, et al. Hawq-v2: Hessian aware trace-weighted quantization of neural networks[J]. Advances in neural information processing systems.
>
> [4] Yao Z, Dong Z, Zheng Z, et al. Hawq-v3: Dyadic neural network quantization[C]//International Conference on Machine Learning. PMLR.
>
> [5] Gui, Shupeng, et al. "Model Compression with Adversarial Robustness: A Unified Optimization Framework." Advances in Neural Information Processing Systems. NeurIPS, 2019.
>
> [6] Lin, Haowen, et al. "Integer-arithmetic-only Certified Robustness for Quantized Neural Networks." Proceedings of the IEEE/CVF International Conference on Computer Vision. ICCV, 2021.

---

### Official Review · Reviewer_tKF2 · 2024-11-03

**Soundness:** 2
**Presentation:** 2
**Contribution:** 2
**Rating:** 3
**Confidence:** 4

**Summary:**

The paper introduces ARQ, a novel mixed-precision quantization framework for executing deep neural networks (DNNs) on resource-constrained platforms while maintaining accuracy and certified robustness. ARQ employs reinforcement learning to find quantization policies that preserve a DNN's accuracy, enhance its robustness, and reduce computational costs, effectively utilizing random smoothing for guidance. The framework supports mixed-precision quantization, allowing different bit-widths for weights in each layer, providing fine-grained control over quantization policies. ARQ is the first to optimize for certified robustness in DNNs and includes random smoothing within its reinforcement learning loop. Experimental results on CIFAR-10 and ImageNet datasets demonstrate that ARQ outperforms state-of-the-art quantization techniques, often matching or exceeding the performance of original FP32 networks with significantly reduced operations. The paper acknowledges ARQ's limitations, such as dependence on the training of the original network and challenges in deploying mixed-precision inference, and suggests future work on applying ARQ to tasks beyond image classification.

**Strengths:**

The paper is easy to read.

**Weaknesses:**

1.	The performance of ARQ is heavily reliant on the quality of the training of the original DNN. If the original network is not properly trained or lacks Gaussian augmentation, the quantized network may not meet expected performance.
2.	ARQ has only been evaluated on image classification tasks. Its performance and effectiveness on other types of tasks have not been validated.
3.	While ARQ performs well in the experiments, its generalizability across different network architectures and datasets remains an open question and requires further research for validation.
4.	The performance of ARQ may be sensitive to hyperparameter choices, which could require additional tuning efforts to ensure optimal performance across different networks and datasets.
5. The presentation of the paper is poor. For examples, the statement of robustness is not clear, adversarial inputs or what? If it is adversarial inputs, why not provide the details for adversarial methods? PGD or FGSM?
6. The method is not novel, and too complex. The results are also not convincing.
7. The main limitations are experiments, i.e., more datasets (COCO2017, VOC), more larger networks (e.g., VIT, GPT-2, Efficientnet), more tasks (detection, segmentation, VQA), and more validation for robustness.
8. Why not compare with more competitors[1-3]?
[1] SDQ: Stochastic Differentiable Quantization with Mixed Precision
[2] EMQ: Evolving Training-free Proxies for Automated Mixed Precision Quantization
[3] OMPQ: Orthogonal Mixed Precision Quantization

**Questions:**

1. What is the definition of robustness for MPQ?
2. What is certified robustness?
3. In your paper, you argue that traditional quantization methods have primarily concentrated on maintaining neural network accuracy, this is not accurate. The goal of MPQ is to study how to reduce the network's parameters, namely, obtaining the trade-off between accuracy and complexity. They have at least two objectives to learn.
4. Why use reinforcement learning?
5. In your paper, you state two objectives, i.e.,  accuracy, and robustness. Why not consider complexity? Maybe you don't understand the MPQ.

---

> ### Author Response · Authors · 2024-11-25
> **Rebuttal to Reviewer tKF2 part 1**
>
> 1.
> > The performance of ARQ is heavily reliant on the quality of the training of the original DNN. If the original network is not properly trained or lacks Gaussian augmentation, the quantized network may not meet expected performance.
>
> > ARQ has only been evaluated on image classification tasks. Its performance and effectiveness on other types of tasks have not been validated.
>
> These limitations were acknowledged in our paper. We plan to explore methods to improve ARQ's robustness under different initial conditions and extend our research to other tasks beyond image classification in future work.
>
> 2.
> > While ARQ performs well in the experiments, its generalizability across different network architectures and datasets remains an open question and requires further research for validation.
> > The main limitations are experiments, i.e., more datasets (COCO2017, VOC), more larger networks (e.g., VIT, GPT-2, Efficientnet), more tasks (detection, segmentation, VQA), and more validation for robustness.
>
> The concerns about generalizability are noted, but it's important to highlight that previous works the reviewer mentioned [1, 2, 3] and the baselines we compared against [4, 5] also primarily focus on ImageNet and ResNet/MobileNet architectures. Our work focuses on the certified robustness, which provides formal guarantees and eliminates necessity for empirical validation.
>
> 3.
> > The performance of ARQ may be sensitive to hyperparameter choices, which could require additional tuning efforts to ensure optimal performance across different networks and datasets.
>
> Section 4 of our paper thoroughly details the hyperparameters used and provides clear instructions for reproducing our results. While hyperparameter sensitivity is inherent in many machine learning methods, we have ensured that our methodology is transparent and reproducible.
>
> 4.
> > The presentation of the paper is poor. For example, the statement of robustness is not clear, adversarial inputs or what? If it is adversarial inputs, why not provide the details for adversarial methods? PGD or FGSM?
> > What is certified robustness?
>
> While other reviewers have noted strengths in our presentation, we are committed to addressing all concerns to improve the clarity and quality of our paper.
> The whole Section 2.2 defines and describes certified robustness. Certified robustness refers to a formal guarantee that a neural network will perform consistently and accurately under perturbations, such as adversarial attacks or input noise.
>
> 5.
> > The method is not novel, and too complex. The results are also not convincing.
>
> We are the first to introduce a mixed-precision quantization framework that optimizes both DNN's accuracy and certified robustness while limiting computational resources. Our approach uniquely uses direct feedback from the ACR of the quantized smoothed classifier. Our experiments demonstrate that ARQ consistently outperforms state-of-the-art quantization methods, often matching or surpassing FP32 networks' accuracy and robustness with just 0.84% of operations. And we show that reinforcement learning can be effectively utilized for such optimization problems.
>
> 6.
> > Why not compare with more competitors[1-3]? [1] SDQ: Stochastic Differentiable Quantization with Mixed Precision [2] EMQ: Evolving Training-free Proxies for Automated Mixed Precision Quantization [3] OMPQ: Orthogonal Mixed Precision Quantization
>
> Firstly, it's important to clarify that the mentioned mixed-precision quantization works [1, 2, 3] do not achieve reasonable robustness without modifications, making them unsuitable as direct competitors. Secondly, we have conducted additional experiments on [7, 8] as requested by reviewers L6Pe and dzkR. ARQ consistently outperforms these baselines, and the detailed results are included in the updated version of our paper.
>
> 7.
> > What is the definition of robustness for MPQ?
>
> In the context of MPQ, robustness is the ability of a quantized neural network to maintain consistent and correct classifications in the presence of adversarial input perturbations. Our approach specifically focuses on certified robustness.

---

> ### Author Response · Authors · 2024-11-25
> **Rebuttal to Reviewer tKF2 part 2**
>
> 8.
> > Why use reinforcement learning?
>
> Previous works [4, 6] have demonstrated the ability of reinforcement learning to efficiently explore and exploit the vast search space on mixed-precision quantization optimization tasks. Our experiments further demonstrate its effectiveness in handling more complex tasks, especially when certified robustness is taken into consideration.
>
> 9.
> > In your paper, you argue that traditional quantization methods have primarily concentrated on maintaining neural network accuracy, this is not accurate. The goal of MPQ is to study how to reduce the network's parameters, namely, obtaining the trade-off between accuracy and complexity. They have at least two objectives to learn.
>
> In Section 2.1, we have clearly stated the goal of quantization method in the first sentence:
> > Neural network quantization is a model compression technique that can reduce a network’s size and compute cost.
>
> We also elaborate on this in Formula 1:
>
> > $Cost(f_P)<C_0$
>
> Additionally, we explain what constitutes the costs:
>
> > $Cost(f_P)$ is the resource usage of the network, such as model size, the number of compute bit operations, or energy consumption, and C0 is a user-specified bound on the resource.
>
> 10.
> > In your paper, you state two objectives, i.e., accuracy, and robustness. Why not consider complexity? Maybe you don't understand the MPQ.
>
>
>
> In our paper, we frame the problem as a multiobjective optimization task. We treat complexity, specifically in terms of bit operations (BOPs), as a constraint. This approach is consistent with the epsilon-method for solving multiobjective problems, where one objective is converted into a constraint to explore solutions across multiple values.
>
>
> Figure 1 clearly illustrates the tradeoff curve between BOPs and ACR/Accuracy, demonstrating how our framework effectively balances these factors. Additionally, Figure 3 in Appendix A.4 provides further insights into the tradeoff between model size and ACR/Accuracy, as requested by Reviewer dzkR.
>
> [1] Huang, Xijie, et al. "SDQ: Stochastic Differentiable Quantization with Mixed Precision." International Conference on Machine Learning (ICML), 2022.
>
> [2] Dong, Peijie, et al. "EMQ: Evolving Training-free Proxies for Automated Mixed Precision Quantization." International Conference on Computer Vision (ICCV), 2023.
>
>
> [3] Ma, Y., et al. "OMPQ: Orthogonal Mixed Precision Quantization." Proceedings of the AAAI Conference on Artificial Intelligence, 37(7), 2023, pp. 9029-9037.
>
> [4] Wang, Kuan, et al. "HAQ: Hardware-Aware Automated Quantization With Mixed Precision." IEEE Conference on Computer Vision and Pattern Recognition (CVPR), 2019.
>
> [5] Tang C, Ouyang K, Wang Z, Zhu Y, Wang Y, Ji W, Zhu W. Mixed-Precision Neural Network Quantization via Learned Layer-wise Importance. arXiv preprint arXiv:2203.08368, 2023.
>
> [6] Lou, Qian, et al. "AutoQ: Automated Kernel-Wise Neural Network Quantization." 8th International Conference on Learning Representations (ICLR), 2020.

---

> > ### Comment · Reviewer_tKF2 · 2024-11-26
> >
> > Thanks for your reply. After reading other reviews and the rebuttal, I hold my score. Your responses are so poor,  the concerns of all reviewers aim to enhance the contribution and clarity of your paper, you should address those concerns immediately. if you ignore now, or plan to address those concerns in the future, why should the reviewers trust that these issues will be resolved later?

---

> > > ### Author Response · Authors · 2024-11-27
> > >
> > > We believe there may be a misunderstanding. We have already extensively updated the paper based on the reviewers' feedback:
> > > * To address the concern about insufficient baseline comparisons, we have conducted additional experiments on four baselines as requested by the reviewers: The results for NIPQ and HAWQ are presented in Figure 1 and Table 1 of the updated paper.
> > > * And the results for ATMC and ICR are shown in Table 8 and Table 9 in Appendix A.6 of updated paper.
> > > * We have also included additional experiments measuring model size, which can be found in Table 1 and Figure 3 in Appendix A.4.
> > > * The experiments addressing the concern about time consumption are discussed in Appendix A.5.
> > > * Regarding your concern that the statement about robustness is unclear, Section 2.2 defines and describes certified robustness in a precise way that is standard in the community.
> > > * For the comment that we overlooked complexity as part of the optimization objective, Section 2.1 explicitly states in the opening sentence that the goal of quantization is to reduce computational cost.
> > > * Additionally, we addressed the complexity cost constraint in Formulas 1, 6, 8 and in Algorithm 1.
> > > * We have also added explanations for the functions used in the algorithm in Section 3.2, as pointed out by reviewer L6Pe.
> > >
> > > Could you please be more precise which questions you still consider unresolved?

---

### Official Review · Reviewer_j8qg · 2024-11-04

**Soundness:** 4
**Presentation:** 4
**Contribution:** 4
**Rating:** 8
**Confidence:** 2

**Summary:**

The paper introduces ARQ (Accurate and Robust Quantization), a framework that addresses the critical challenge of deploying deep neural networks in resource-constrained environments while maintaining both accuracy and robustness. The framework's key innovation lies in its novel approach to mixed-precision quantization that directly optimizes for certified robustness, making it the first of its kind in the field. Through a sophisticated combination of reinforcement learning and randomized smoothing, ARQ achieves remarkable efficiency gains while preserving or even improving model performance compared to full-precision networks.

**Strengths:**

The use of average certified radius as an optimization objective is particularly novel, allowing simultaneous optimization of accuracy and robustness.
The quality of the research is evident in the comprehensive experimental validation across multiple architectures and datasets. The authors provide ablation studies and comparative analysis which gives valuable insights into the framework's behavior and advantages. The achievement of matching or exceeding full-precision performance with drastically reduced computational requirements is particularly impressive.

**Weaknesses:**

While the paper's contributions are substantial, there are areas that could benefit from further exploration. The current focus on convolutional neural networks and image classification, while well-executed, leaves open questions about generalization to other architectures and tasks. This limitation is acknowledged by the authors, but additional discussion of potential adaptation strategies would be valuable.
The paper would benefit from more detailed analysis of memory usage patterns during training, though the runtime analysis provided in Table 2 is helpful. Additionally, while the framework's theoretical foundations are strong, more discussion of practical hardware implementation considerations for mixed-precision inference would strengthen the work's immediate applicability.

**Questions:**

How might the ARQ framework be adapted for sequence models or transformers, particularly regarding the interaction between attention mechanisms and mixed-precision quantization?
What modifications to the incremental randomized smoothing approach might be necessary to support other certification methods while maintaining computational efficiency?
How does the framework handle the transition between different precision levels during inference, and what are the implications for hardware implementation?
Could the reinforcement learning approach be extended to dynamically adjust precision levels during inference based on input characteristics?

---

> ### Author Response · Authors · 2024-12-04
>
> We thank reviewer 32H8 for recognizing the importance of our work and for the insightful suggestions. We respond to each of these below.
>
> > The paper would benefit from more detailed analysis of memory usage patterns during training, though the runtime analysis provided in Table 2 is helpful.
>
> Thank you for acknowledging our runtime analysis! Below is a table that analyzes the memory usage of different models during training:
> | Model | Memory Usage (MiB) |
> |---------------|--------------------|
> | ResNet-20 | 2150 MiB |
> | ResNet-50 | 15412 MiB |
> | MobileNet-V2 | 12490 MiB |
>
> The memory usage is closely tied to the number of parameters in the models and the input batch size. Training or fine-tuning models typically requires a similar amount of memory usage, so the observed memory consumption aligns with standard expectations for such tasks. Users can optimize memory consumption by adjusting the batch size during fine-tuning and robustness certification.
>
> > More discussion of practical hardware implementation considerations for mixed-precision inference would strengthen the work's immediate applicability.
>
> > How does the framework handle the transition between different precision levels during inference, and what are the implications for hardware implementation?
>
> Previous work Bit Fushion [5] introduced bit-level composable architectures that can dynamically adjust to the required bit-widths. This approach allows the architecture to dynamically adjust the bit-widths of operations to match the specific precision requirements of each layer in a neural network.
>
> The architecture in [5] can reconfigure itself at inference to accommodate varying bit-widths required by different layers. [5] uses an array of bit-level processing elements that can fuse or decompose to match the bit-width of individual DNN layers. The ARQ framework can integrate well into this architecture because ARQ assigns different bit-widths on a per-layer basis and employs linear quantization to simplify deployment. This compatibility allows ARQ to utilize Bit Fusion [5]'s dynamic bit-level composability, ensuring that each layer operates at its optimal precision.
>
> > How might the ARQ framework be adapted for sequence models or transformers, particularly regarding the interaction between attention mechanisms and mixed-precision quantization?
>
> This is a good question! Previous works [1, 2] have explored the application of randomized smoothing to sequence models and transformers. [3, 4] have provided solutions of mixed-precision quantization on sequence models and transformers. These works can be helpful for extending the ARQ framework to NLP tasks. We are willing to explore these further in our future work.
>
> > What modifications to the incremental randomized smoothing approach might be necessary to support other certification methods while maintaining computational efficiency?
>
>
> Incremental randomized smoothing is designed for randomized smoothing’s verification process by estimating the disparity ζx– the upper bound on the probability that outputs of smooth classifier f and quantized smooth classifier f^p are distinct. To broaden its applicability, the core algorithm should be restructured to align with the principles of new certification methods. This might involve redefining the disparity estimation process and adjusting algorithmic steps to accommodate different verification methods.
>
> > Could the reinforcement learning approach be extended to dynamically adjust precision levels during inference based on input characteristics?
>
> That is a great idea! It's indeed possible to extend the reinforcement learning approach to dynamically adjust precision levels during inference based on input characteristics. To achieve this, we could consider increasing the quantization bit-level to a range that does not require additional fine-tuning. Given the high demand for certification (e.g., 100,000 noisy samples for a single image), it’s possible to use a subset of the noisy samples to determine optimal quantization bit-levels and then use them for further certification.
>
> [1] Moon, Han Cheol, et al. "Randomized Smoothing with Masked Inference for Adversarially Robust Text Classifications." In Proceedings of the 61st Annual Meeting of the Association for Computational Linguistics (ACL), 2023.
>
>
> [2] von Oswald, Johannes, et al.. "Learning Randomized Algorithms with Transformers." arXiv preprint arXiv:2408.10818, 2024.
>
> [3] Liu, Zhenhua, et al. "Post-training quantization for vision transformer." Advances in Neural Information Processing Systems 34 (2021): 28092-28103.
>
>
> [4] Yao, Zhewei, et al. "Zeroquant: Efficient and affordable post-training quantization for large-scale transformers." Advances in Neural Information Processing Systems 35 (2022): 27168-27183.
>
> [5] H. Sharma et al., "Bit Fusion: Bit-Level Dynamically Composable Architecture for Accelerating Deep Neural Network," 2018 ACM/IEEE 45th Annual International Symposium on Computer Architecture (ISCA)

---

### Meta-Review · Area_Chair_YjDJ · 2024-12-20

**Metareview:**

This paper introduces ARQ, a mixed-precision quantization (MPQ) framework that optimizes accuracy and certified robustness for deep neural networks (DNNs) under resource constraints. ARQ employs reinforcement learning to determine optimal bit-width assignments across layers, with incremental randomized smoothing guiding the optimization process to maximize the average certified radius (ACR). The framework demonstrates strong results, matching or exceeding the performance of full-precision models while significantly reducing computational requirements. Reviewers commend ARQ's motivation, clear presentation, and novel focus on robustness in MPQ, particularly the integration of robustness certification into the quantization process. However, concerns are raised regarding limited experimental validation beyond image classification tasks, weak comparisons with advanced MPQ methods, and the lack of evaluations on more diverse datasets, architectures, and tasks. Additionally, the reliance on reinforcement learning and the absence of clarity on specific algorithmic details are criticized. While the contributions are notable, the incremental nature of the method and insufficient comparisons with state-of-the-art techniques leave the paper below the acceptance threshold for most reviewers.

**Additional Comments On Reviewer Discussion:**

The reviewer criticizes the authors for failing to adequately address the issues and questions the trustworthiness of their commitment to resolving these problems in the future. The authors highlight additional experiments conducted on four baselines (e.g., NIPQ, HAWQ, ATMC, and ICR), added results on model size, and addressed concerns about time consumption and robustness definitions through updated sections, tables, and appendices. Since major reviewers raised negative scores, the meta-reviewer encourages the authors to polish the paper according to the reviews.

---

### Decision · Program_Chairs · 2025-01-22

Reject